# Effective Activation of Human Antigen-Presenting Cells and Cytotoxic CD8^+^ T Cells by a Calcium Phosphate-Based Nanoparticle Vaccine Delivery System

**DOI:** 10.3390/vaccines8010110

**Published:** 2020-03-01

**Authors:** Florian Scheffel, Torben Knuschke, Lucas Otto, Sebastian Kollenda, Viktoriya Sokolova, Christine Cosmovici, Jan Buer, Jörg Timm, Matthias Epple, Astrid M. Westendorf

**Affiliations:** 1Institute of Medical Microbiology, University of Duisburg-Essen, 45122 Essen, Germany; 2Institute for Virology, University of Duisburg-Essen, 45122 Essen, Germany; 3Institute for Experimental Immunology and Imaging, University of Duisburg-Essen, 45122 Essen, Germany; 4Inorganic Chemistry and Center for Nanointegration Duisburg-Essen (CeNIDE), University of Duisburg-Essen, 45122 Essen, Germanyviktoriya.sokolova@uni-due.de (V.S.);; 5Institute of Virology, Heinrich-Heine-University, Medical Faculty, 40225 Düsseldorf, Germany

**Keywords:** nanoparticles, vaccine, cytotoxic T cells, immune modulation, immunotherapy

## Abstract

The ability of vaccines to induce T cell responses is crucial for preventing diseases caused by viruses. Nanoparticles (NPs) are considered to be efficient tools for the initiation of potent immune responses. Calcium phosphate (CaP) NPs are a class of biodegradable nanocarriers that are able to deliver immune activating molecules across physiological barriers. Therefore, the aim of this study was to assess whether Toll-like receptor (TLR) ligand and viral antigen functionalized CaP NPs are capable of inducing efficient maturation of human antigen presenting cells (APC). To achieve this, we generated primary human dendritic cells (DCs) and stimulated them with CpG or poly(I:C) functionalized CaP NPs. DCs were profoundly stronger when activated upon NP stimulation compared to treatment with soluble TLR ligands. This is indicated by increased levels of costimulatory molecules and the secretion of proinflammatory cytokines. Consequently, coculture of NP-stimulated APCs with CD8^+^ T cells resulted in a significant expansion of virus-specific T cells. In summary, our data suggest that functionalized CaP NPs are a suitable tool for activating human virus-specific CD8^+^ T cells and may represent an excellent vaccine delivery system.

## 1. Introduction

Chronic viral diseases and cancer are associated with immune suppression, i.e., impaired antigen presentation or T cell dysfunction [1]. Several immunotherapeutic strategies are under investigation to overcome the hurdles for successful treatment of infected patients suffering from these diseases. Nanoparticle (NP)-based carrier platforms facilitating the delivery of biomolecules are increasingly under investigation [2,3,4]. Due to their small size, nanoparticles (NPs) are predestined to target and to be engulfed by antigen presenting cells (APCs) such as dendritic cells (DC) or macrophages [5]. However, their size, their surface charge, and their surface functionalization strongly affect the uptake by those cells. Particles with a size of 20–100 nm can directly enter into the lymphatic vessels reaching peripheral lymph nodes, where they are captured mostly by DCs [6]. DCs are the most potent APCs, which expresses subtype dependent Toll-like receptors (TLR) and interact with CD4^+^ and CD8^+^ T cells by processing and presenting antigens on major histocompatibility complex (MHC) class II or I molecules [7]. Different DC subsets have been identified, which are variously located in the human body and are specialized to regulate specific immune responses to pathogens or danger signals [8]. While plasmacytoid DCs (pDC) are mainly found in the blood, the majority of conventional DCs (cDC) are found in lymphoid tissues and organs, e.g., lung, intestine, and liver. cDCs can be further subdivided into the two subsets cDC1 and cDC2 previously defined by CD141 and CD1c expression [9]. pDCs express the intracellular TLRs 7 and 9, while cDCs respond through TLRs 1-6, 8, and 10 to lipopolysaccharide, flagellin, poly(I:C), and R848 [10,11]. Activation of these conserved pathogen recognizing receptors by their ligands induces maturation of DCs including the secretion of pro-inflammatory cytokines, which orchestrate the immune response. Thus, an ideal delivery vehicle should actively or passively target DCs and protect the antigenic material as well as TLR ligands to induce highly efficient DC maturation. Calcium phosphate (CaP) is an inorganic component of many biological hard tissues such as bone and teeth [12,13]. Therefore, it shows a high biocompatibility and biodegradability, which is a clear advantage compared to other compounds used for nanoparticle production. CaP is an important agent in various fields of biomedicine, especially for the treatment of bone defects but has also been added to cosmetics [14,15]. Moreover, colloidal stabilization of CaP NPs with suitable agents enables their use as a particulate carrier system to transport drugs and biomolecules into cells. CaP NPs for vaccination purposes are phagocytosed by DCs and, subsequently, dissolved in lysosomes [16]. Previously, we have established the synthesis and use of the TLR ligand and antigen functionalized CaP NPs in murine chronic viral infection and tumor models [17,18]. Cytotoxic T lymphocytes (CTL) are known to be crucial for the elimination of virus infected cells or tumor cells. We showed that NP-based vaccination of chronically infected mice leads to strong reinforcement of cytotoxic CD8^+^ T cells and a significant reduction of the viral load [19]. This demonstrates the potential efficacy of CaP nanoparticles as a therapeutic vaccine delivery system. For example, until today, there is no effective immune therapeutic approach available to treat HIV-infected or HBV-infected patients underlining the need for new potential approaches.

Importantly, so far, no studies focused on the efficacy of the CaP nanoparticle system for the activation of human cells. In the present study, we used a cytomegalovirus (CMV) derived peptide as a model antigen and poly(I:C) or CpG functionalized CaP NPs to activate peripheral blood mononuclear cell (PBMC)-derived primary DCs as well as monocyte-derived DCs (mDC). Our data demonstrate, for the first time, the high potential of CaP NPs as a vaccination tool for human purposes, targeting different subsets of human DCs, and leading to a potent virus-specific CD8^+^ T cell response.

## 2. Materials and Methods

### 2.1. TLR-Ligands and Viral Peptides

The phosphorothioate-modified class C CpG ODN M362 and polyinosinic-polycytidylic acid potassium salt (poly(I:C)) were purchased from Sigma Aldrich. The CMV pp65 derived HLA-A*02-restricted epitope NLVPMVATV was synthesized (EMC microcollections, Tübingen, Germany). 

For uptake studies, green fluorescing CpG labeled with Alexa488 (Life Technologies, Carlsbad, USA) was used for the preparation of triple-shell NPs (CaP/CpGAlexa488/CaP/CpGAlexa488), as described below.

### 2.2. Synthesis of Functionalized CaP Nanoparticles

Triple-shell CaP nanoparticles were prepared as described previously [20]. According to this synthesis, nanoparticles loaded with CMV pp65 peptide were prepared. Briefly, 0.5 mL of calcium nitrate solution (6.25 mM), 0.5 mL of diammonium hydrogen phosphate solution (3.74 mM), and 0.1 mL CpG, and/or poly(I:C) were mixed together to form single-shell nanoparticles. Additionally, 50 µL of the cytomegalo virus (CMV) peptide (1 mg/mL) followed by another 0.5 mL of calcium nitrate solution (6.25 mM) and 0.5 mL of diammonium hydrogen phosphate solution (3.74 mM) were added to the dispersion of single-shell nanoparticles, i.e., CaP/CpG or CaP/poly(I:C), which leads to the triple-shell nanoparticles CaP/CpG/CMV_pp65_/CaP/CpG, CaP/poly(I:C)/CMV_pp65_/CaP/poly(I:C), and CaP/CpG/poly(I:C)/CMV_pp65_/CaP/CpG/poly(I:C)), respectively.

Next, the triple-shell nanoparticles were removed from the supernatant by centrifugation for 15 min at 12,000 g (MiniSpin centrifuge, Eppendorf, Wesseling, Germany). The concentration of the incorporated biomolecules was determined by UV/Vis spectroscopy on the supernatant in a DS-11 FX+ Nanodrop instrument (DeNovix, Wilmington, USA). The final concentrations of CpG, poly(I:C), and the CMV peptide in the particle dispersions were 20.6 µM (125 µg/mL), 167 mg/mL, and 41 µg/mL, respectively. Then, the centrifuged NPs were re-dispersed in water (typically in 1 mL) under ultrasonication (Elmasonic S10 water bath, Elma, Singen, Germany) for 10 s.

All inorganic salts were of *pro analysi* (p.a.) quality. Ultrapure water (Purelab ultra instrument from ELGA, Celle, Germany) was used for all preparations. All formulations were prepared and analyzed at room temperature and sterile-filtered before application. The particles were characterized by scanning electron microscopy (ESEM Quanta 400, FEI, Oregon USA) with palladium-gold sputtered samples. Dynamic light scattering was performed with a Zetasizer nanoseries instrument (Malvern Nano ZS, λ = 532 nm). The particle size data refer to scattering number distributions. Calcium concentrations were determined by atomic absorption spectroscopy (AAS, M-Series AA spectrometer, Thermo Electron Corporation, Schwerte, Germany) after dissolution of the particles in hydrochloric acid.

### 2.3. Subjects

HLA-A*0201+ CMV positive individuals were included in this study, with approval by the Ethics Committee of the Medical Faculty of the Heinrich-Heine-University Düsseldorf (2018-131-KFogU).

### 2.4. Isolation of PBMCs from Human Blood

Blood mononuclear cells (PBMCs) were separated from whole blood by the use of Ficoll (Roth, Karlsruhe, Germany) and Leucosep centrifuge tubes (Greiner Bio-One, Frickenhausen, Germany).

### 2.5. Isolation of Primary DCs

cDCs were isolated by using the Blood Dendritic Cell Isolation Kit II (Miltenyi Biotech, Bergisch Gladbach, Germany). For the isolation of pDCs, the Diamond Plasmacytoid Dendritic Cell Isolation Kit II (Miltenyi Biotech) was used. According to the manufacturer’s instructions, cells were purified via an AutoMACS pro separator.

### 2.6. Generation of Monocyte-Derived DCs

PBMCs from CMV-positive donors were thawed in IMDM medium (Gibco, Schwerte, Germany) containing 5% penicillin/streptomycin (ThermoScientific, Waltham, USA). After centrifugation at 630 g for 3 min, the cells were resuspended in DC-medium (IMDM medium supplemented with 100 µg/mL penicillin/streptomycin and 1% human serum (Pan Biotech)). After 3 h of incubation at 37 °C and 5% of CO_2_, the cells were rinsed and the medium was replaced with 3 mL of fresh DC-medium supplemented with 800 U/mL rhGM-CSF (Peprotech, Hamburg, USA) and 1000 U/mL rhIL-4 (Peprotech,). The cells were then incubated for 48 h at 37 °C 5% CO_2_. Lastly, 1.5 mL of DC-medium containing rhGM-CSF and rhIL-4 was added. After another 24 h of incubation, mDCs were harvested and generation of mDCs was verified by flow cytometry staining for CD11c^+^ CD14^−^ cells.

### 2.7. Uptake of Alexa488-Labeled CaP Nanoparticles by DCs

For the demonstration of nanoparticle uptake by confocal laser scanning microscopy (CSLM), cDCs were isolated as described and seeded in a Nunc™ Lab-Tek™ II 4-Chamber Slide™ System (ThermoFisher, Waltham, USA) with a density of 250,000 cells per well in 500 µL medium. After that, 10 µL of the nanoparticle dispersions (CaP/CpG488/CaP/CpG488) were added to the cells. After 3 h and 24 h, the whole cell culture medium was removed with a pipette and the cells were washed with Dulbecco’s phosphate buffered saline (PBS) and fixed with 4% paraformaldehyde solution at room temperature for 10 min, which was followed by washing for three times with DPBS. Actin staining was done with Alexa Fluor 647 according to the manufacturer’s procedure (Thermo Fischer), which was followed by washing for three times with DPBS. For labelling nuclei, the cells were then stained with DAPI and washed three times with DPBS. After staining, the samples were transferred to the slide using Fluorophore G mounting media. Lastly, the cells were studied by CLSM. To identify the location of the nanoparticles within the cells, z-stacks were acquired (step size 0.3 µm). 1 × 10^6^ PBMCs per well were stimulated with 40 µL of a nanoparticle dispersion containing approximately 1.29 × 10^10^ Alexa488-labeled CpG functionalized CaP NPs for 3, 6, 16, and 24 h in 400 µL of IMDM-complemented medium containing 10% FCS, 25 µmol 2-β-mercaptoethanol, and 100 µg/mL penicillin/streptomycin in a 48-well plate at 37 °C. NP-positive DCs were identified by flow cytometric analysis for CD11c^+^, CD14^-^, Alexa488^+^ events.

CLSM was performed with a TCS SP8 AOBS system running LASX software (Leica Microsystems, Wetzlar, Germany). The laser lines used for excitation were Diode 405 nm (DAPI, detection range 410-460 nm), Argon 488 nm (CpG488, detection range 488–520 nm), and HeNe 633 nm (Alexa Fluor 647, detection range 640–720 nm). Images were acquired with an HCX PL apo 63x/.4 oil objective.

### 2.8. Stimulation of DCs with Functionalized CaP Nanoparticles

A total of 5 × 10^5^ isolated cDCs or pDCs per well were stimulated with 40 µL of the CMV_pp65_ peptide and soluble TLR ligands CpG, poly(I:C) or CaP NPs containing the peptide and these TLR ligands in 400 µL of complemented IMDM in a 48-well plate at 37 °C. Particle concentrations were 1.29 × 10^10^ CpG functionalized CaP NPs and 1.748 × 10^10^ poly(I:C) functionalized CaP NPs per well. After 24 h, supernatants and cells were harvested and used for further analysis.

mDCs were harvested and adjusted to 3 × 10^5^ cells/mL on a 12-well plate. To activate the mDCs, 20 µL of PBS, CMV_pp65_ peptide, and soluble TLR ligands or a combination of both or TLR ligand functionalized CaP NPs was added. Particle concentrations were 6.45 × 10^9^ CpG functionalized CaP NPs, 8.74 × 10^9^ poly(I:C) functionalized CaP NPs, and 4.52 × 10^8^ CpG/poly(I:C) functionalized CaP NPs per well. After incubation for 24 h, the supernatant was collected for cytokine analysis. To resuspend the cells, 2 mL of PBS containing 1% human serum was added per well. The cells were incubated in the fridge for 30 min to detach them from the plate. Harvested mDCs were then analyzed via flow cytometry or further used for co-culturing with CD8^+^ T cells.

### 2.9. Isolation of CD8^+^ T Cells

CD8^+^ T cells were isolated using the CD8 negative selection kit (Miltenyi Biotech) according to the manufacturer’s instructions via an AutoMACS pro seperator. The purity of the isolated cells was verified by flow cytometry and was consistently between 75% and 90%.

### 2.10. In vitro Co-Culture of NP-Stimulated DCs and Virus-Specific CD8^+^ T Cells

After activation, the mDCs were examined by microscopy for plastic adherent, elongated cells. Small, non-adherent cells were discarded and PBS with 1% human serum was added to detach the cells. After incubation at 4 °C for 30 min, the cells were harvested and adjusted with DC-medium to 50,000 cells/100 µL. Isolated CD8^+^ T cells were adjusted with T cell-medium (IMDM + 5% PenStrep + 5% human serum) to the same concentration as the mDCs. mDCs and T cells were then cultivated in a 48-well-plate with a ratio of 1:4 at 37 °C, 5% CO_2_ for three days or seven days. The experiment was performed six times using cells from a different donor for each experiment.

For re-stimulation of CD8^+^ T cells, peptide-pulsed blood cells of the same donor were used as target cells. Therefore, 1 × 10^7^ PBMCs were incubated in 6 mL of DC-medium supplemented with a 1 µg/mL CMV peptide for 1 h at 37 °C, 5% CO_2_ followed by two washing steps, and resuspension in DC-medium. CD8^+^ T cells were mixed with 2x10^5^ peptide-pulsed target cells in a 96-well-plate and incubated for 5 h in the presence of 0.2% Brefeldin A (Sigma Aldrich, Darmstadt, Germany). Intracellular staining for IFNγ was performed afterward by following a standard protocol.

### 2.11. Antibodies and Flow Cytometry

The monoclonal antibodies αCD11c (clone B-ly6) and αCD14 (clone M5E2) were purchased from BD Biosciences Pharmingen. αCD8 (clone MEM-31) monoclonal antibody was obtained from Immuno Tools. αCD80 (clone 2D10), αCD86 (clone IT2.2), and αIFNγ (clone B27) antibodies were used from Biolegend. The αKi67 (clone SoIA15) antibody was purchased from ebioscience. Intracellular staining with αIFNγ and αKi67 was performed as described previously [17]. Data were acquired by using an LSR II instrument using DIVA software version 6 (BD Biosciences Pharmingen).

### 2.12. Cytokine Detection in Supernatants

DCs were stimulated with functionalized CaP NPs, as previously described. Cytokines in the cell culture supernatants of primary DCs were measured by polystyrene bead-based Luminex technology (R&D Systems), according to the manufacturer’s instructions. The sample was mixed with labeled beads and pre-coated with analyte-specific antibodies. Beads were read on a dual-laser flow-based detection instrument. A Luminex200 instrument (Luminex Corporation, Austin, USA) was used to run the assay. Cytokines in the cell culture supernatants of mDCs were measured by magnetic bead-based Luminex technology, according to the manufacturer’s instructions (R&D Systems). A MAGPIX instrument (Luminex Corporation) was used for evaluation.

## 3. Results

### 3.1. Colloid-Chemical Characterization of Functionalized Calcium Phosphate Nanoparticles

Scanning electron microscopy (SEM) showed spherical NPs with a typical diameter of 60 to 75 nm (Figure 1). Dynamic light scattering gave a hydrodynamic radius of 100 nm. The negative zeta potential of −22 mV was due to the outer shell of anionic CpG or poly(I:C). The calcium concentration in the dispersions was 58 μg/mL Ca^2+^ for the CpG-carrying particles and 62.2 μg/mL Ca^2+^ for the poly(I:C)-carrying particles and 63 μg/mL Ca^2+^ for the CpG/poly(I:C)-carrying particles (by AAS), which results in estimated calcium phosphate concentrations of 147, 156, and 158 μg/mL, respectively (tentative assumption of the stoichiometry of hydroxyapatite, Ca_5_(PO_4_)_3_OH). Together with a particle diameter from SEM (60–75 nm), this gives a particle concentration of 3.23 × 10^11^, 4.37 × 10^11^, and 2.26 × 10^11^ particles per mL. With these data, the mass ratios of biomolecules (CpG, poly(I:C) and CMV peptide) to CaP nanoparticles were calculated to 0.85: 1 = CpG: CaP, 1.07: 1 = poly(I:C): CaP, 0.79: 1.06: 1 = CpG: poly(I:C): CaP, and 0.28: 1 = peptide: CaP.

### 3.2. Uptake of Functionalized CaP Nanoparticles by Human Antigen Presenting Cells

APCs are specialized phagocytotic cells and are predestined to internalize small molecules such as NPs [5]. DCs have a large impact on the induction of adaptive immune responses [21]. Therefore, targeting APCs with a combined formulation consisting of an adjuvant and relevant antigens is an important goal in the development of potential vaccines against viruses or cancer. To determine whether DCs can take up CaP NPs *in vitro*, cDCs were isolated from peripheral blood and incubated with CpG-Alexa488-labelled CaP NPs for 3 h and 24 h. We were able to detect a distinct increase of NP uptake over time by confocal laser scanning microscopy (CLSM) (Figure 2A). The nanoparticles were taken up by cells but did not enter the nucleus. To confirm these results, PBMCs were stimulated with CpG-Alexa488-labelled CaP NPs and analyzed by flow cytometry. After 3 h of stimulation, ~75% of the cells among the DC (CD11c^+^ CD14^−^) population were positive for Alexa488, which demonstrates a high uptake efficacy (Figure 2B,C). Additionally, 24 h after stimulation, nearly all cells were NP-positive (~97%).

### 3.3. Maturation of Blood Derived DCs by Functionalized CaP Nanoparticles

Different human DC subsets have been identified, where each is specialized to regulate immunity in response to danger signals. While cDCs express with TLR 1-6, 8, and 10, a wide range of TLRs. pDCs only express TLR 7 and 9 [10,11]. Therefore, we functionalized CaP NPs with the TLR 3 ligand poly(I:C) to target cDCs or TLR 9 ligand CpG to induce a potent maturation of pDCs toward mature APCs. Recognition of these ligands induces several phenotypic changes, e.g., the upregulation of the expression of costimulatory surface molecules, antigen presentation, and the expression of proinflammatory cytokines [22]. These processes are mandatory for efficient activation and induction of T cell responses. cDCs and pDCs were purified from peripheral blood mononuclear cells (PBMC) and incubated with poly(I:C) (cDC) or CpG (pDC) functionalized CaP NPs or soluble TLR ligands for 24 h. The uptake of functionalized CaP NPs induced the enhanced expression of the costimulatory molecules CD86 and CD80 on cDC subsets. However, while CD86 was expressed higher on cDCs after CaP NP stimulation compared to soluble poly(I:C), there was no difference in the expression of CD80 (Figure 3A). After 24 h of stimulation, the expression of the MHC class I molecule HLA-ABC and the MHC class II molecule HLA-DR was also enhanced (Figure 3B). We also analyzed the production of proinflammatory cytokines 24 h after stimulation with functionalized CaP NPs. Significantly higher amounts of IL-2, TNFα, IL-12, and the inflammasome cytokine IL-1β were detected in the supernatants of the cDC cultures compared to soluble poly(I:C) stimulation (Figure 3C). In contrast, only low amounts of IFN-α were secreted by cDCs after treatment. Stimulation of pDCs with CpG functionalized CaP NPs led to a significantly increased expression of CD86. However, there was no difference between the stimulation with CaP NPs and a soluble TLR ligand. The expression of CD80 on pDCs after stimulation with soluble CpG or CpG functionalized CaP NPs was only slightly enhanced compared to unstimulated cells (Figure 3D). To a lesser extent, surface expression of HLA-ABC and HLA-DR was also enhanced on pDCs (Figure 3E). As expected, the cytokine expression profile of pDCs differed from that of cDCs. Stimulation with CpG functionalized CaP NPs only induced significantly higher production of TNFα, IL-1β, and IFN-α while IL-2 and IL-12 were not induced by CpG (Figure 3F). Thus, our results demonstrate that stimulation of cDCs and pDCs with different TLR ligand functionalized CaP NPs induce efficient maturation of specific DC subsets.

### 3.4. Maturation of Monocyte Derived DCs by Functionalized CaP Nanoparticles

Since the percentage of DCs in the peripheral blood is rather low, the generation of monocyte derived DCs (mDC) is a widely used technique to boost or induce immune responses as part of adoptive transfer-based immunotherapeutic approaches. These mDCs are potent APCs, and are fully equipped with all the necessary functions to prime T cell responses [23]. To test whether CaP NP matured DCs are capable to activate and expand virus-specific CD8^+^ T cell responses, we first stimulated mDCs generated from monocytes of cytomegalovirus (CMV) positive patients with CpG or poly(I:C) functionalized CaP NPs or both. Additionally, we encapsulated a CMV-derived MHC class I restricted CD8^+^ T cell epitope. As expected, mDCs showed an increased expression of CD80 and CD86 after stimulation with CpG, poly(I:C), or CpG/poly(I:C) functionalized CaP NPs compared to unstimulated cells or to cells after stimulation with soluble TLR ligands (Figure 4A). HLA-ABC and HLA-DR expression after CaP NP stimulation were increased when compared to treatment with soluble TLR ligands. The treatment with NPs containing both CpG and poly(I:C) led to a significantly high expression of HLA-ABC by mDCs compared to unstimulated cells. Concomitantly, these matured cells secrete high levels of TNFα, IL-12, IL-1β, IL-2, and the IFN-I cytokine IFN-α after CaP NP stimulation (Figure 4B). Additionally, TLR ligand functionalized CaP NPs were more effective in inducing the cytokine secretion than the soluble ligands.

Subsequently, we incubated the CaP NP stimulated matured mDCs with CD8^+^ T cells isolated from the blood of CMV-positive patients and analyzed the expansion of CMV-specific CD8^+^ T cells. After three days of co-culturing activated mDCs and CD8^+^ T cells, we noticed a significant increase of CMV-specific CD8^+^ T cells when DCs were stimulated with functionalized CaP NPs (Figure 5A,B). However, the relative percentage of CMV-specific IFNγ^+^ CD8^+^ T cells further increased after seven days of co-culturing when DCs were treated with poly(I:C) or CpG functionalized CaP NPs. Only in the groups where mDCs were treated with functionalized CaP NPs, we detected a significant increase of IFNγ-producing CD8^+^ T cells. To investigate the proliferation of the CD8^+^ T cells, we stained for the presence of the cellular marker for proliferation Ki67. Although there was no significant difference in the proliferation between poly(I:C) or/and CpG functionalized NPs compared to their soluble TLR ligand controls, we detected enhanced percentages of Ki67^+^ CD8^+^ T cells after co-culturing with mDCs that were treated with functionalized NPs (Figure 5C).

## 4. Discussion

In the current study, we demonstrate that TLR ligand and antigen functionalized CaP NPs are capable to induce efficient DC maturation derived from human PBMCs, which results in enhanced antigen-specific CD8^+^ T cell activation. Our results constrain for the first time that the CaP NP carrier system may be beneficial for prophylactic or therapeutic vaccination approaches in humans.

Several strategies have been investigated to increase the maturation and the uptake of antigens by APCs. One possibility is the conjugation of TLR ligands to DC-specific antibodies [24] while another is the conjugation of TLR ligands to antigens [25]. Delivering antigens and adjuvants such as TLR ligands in particulate form facilitates passive targeting to DCs, which enhances antigenicity and activation [26]. For the activation of APCs, we used the TLR ligands CpG and poly(I:C). CpG is an established adjuvant for immunotherapeutic purposes [27]. CpG triggers the endosomal localized TLR 9 expressed at high levels by human pDCs, which leads to type I IFN expression through IRF7 or NFκB mediated expression of proinflammatory cytokines that regulate adaptive immune responses [28]. Poly(I:C) is a synthetic analog of double-stranded RNA and binds to TLR 3, which is selectively expressed by human DCs [29]. While systemic application of TLR ligands is linked with limited success during immunotherapy, the encapsulation into particulate formulations extends the durability and effectiveness by decreasing the necessary dose [30]. We observed similar effects using CpG or poly(I:C) functionalized CaP NPs for the stimulation of cDCs or pDCs isolated from human PBMCs as well as monocyte-derived DCs. Especially, the secretion of proinflammatory cytokines such as IL-12, IL-2, or TNFα was highly increased after CaP NP treatment of cDCs. Stimulation with soluble TLR ligands led to secretion of markedly less levels. This is well in line with our previous results from mouse models and could be due to the enhanced delivery of the ligands to endosomal compartments, where TLR 3 and TLR 9 are located [16,20]. IL-12, IL-2, IFN-α, and TNFα are very important for the sufficient activation of effector CD8^+^ T cells [19,31,32,33] or in an autocrine manner for enhanced DC function [34]. Therefore, this is an optimal prerequisite for effective T cell priming. Even though we found differences for the cytokine secretion of cDCs after NP-poly(I:C) stimulation in comparison to soluble poly(I:C), we did not observe differences in the expression of costimulatory molecules by these cells. This might be due to the time point at which we analyzed the cells. Phenotypic maturation, which is indicated by the upregulation of costimulatory molecules, occurs very early while expression of cytokines needs more time [35]. Therefore, analysis at an earlier time point might show stronger differences regarding costimulatory molecule expression. Concentrations of the TLR ligands were adjusted from in vivo data that was generated previously [17,19]. Lower concentrations of NPs or TLR ligands might also emphasize the advantage of CaP NPs. The induction of a potent CD8^+^ T cell response is very important for the control of viral infections, such as HIV or tumor disease [36]. In this regard, it has been demonstrated that triggering TLRs enhances the T cell priming process to increase the robustness of adaptive immune responses [37]. Encapsulation of these ligands into NPs could further improve this effect. Previously, we described the efficient uptake of CpG functionalized CaP nanoparticles by and activation of DCs in mouse models, which results in enhanced activation of CD8^+^ effector T cell responses [17,18,19]. CD8^+^ T cells gained activation and expression of effector molecules such as IFNγ or granzyme B, which leads to repressed tumor growth or enhanced elimination of virus-infected cells. Importantly, we now demonstrate that CaP NP stimulated human monocyte derived DCs were capable of inducing cytotoxic CD8^+^ T cell responses. CaP NP activated DCs were more efficient in inducing IFNγ-producing virus-specific CD8^+^ T cell than soluble TLR ligands and antigens. This could mean that, although the expression of costimulatory molecules between NP and soluble TLR ligand treated mDCs was not significantly different, the uptake of encapsulated antigens was more efficient as well as the ability to cross present these exogenous antigens by encapsulation. While the expression of the MHC class I molecule was strongly increased on mDCs after stimulation with NPs functionalized with poly(I:C) and CpG, these DCs already induced the highest CD8^+^ T cell response after three days. We were also able to detect proliferation of CD8^+^ T cells. We observed proliferation of CD8^+^ T cells, but the percentages of Ki67^+^ cells within this CD8^+^ T cell pool is low due to the fact that the majority of the T cells in the co-culture is still unspecific for the CMVpp65 peptide. Previous studies have demonstrated that nanoparticle-mediated delivery of antigens can dramatically enhance and sustain exogenous antigen presentation by MHC class I [38,39]. In a previous study, we already demonstrated in vitro that encapsulation of poly(I:C) or/and CpG together with an MHC class II restricted model antigen is suitable for the induction of a CD4^+^ T cell response [20]. Thus, antigens transported via CaP NPs are presented via MHC I and II restricted pathways. This is important since an effective therapeutic vaccine against HIV is still not available. Although antiretroviral therapy (ART) effectively represses viral replication and strongly improves clinical outcome, life-long treatment is required. Abortion of ART will lead to reactivation of HIV from latent reservoirs associated with a high level of replication and progressive disease. Therefore, new strategies are under investigation to eradicate the HIV reservoir. One approach is the “shock and kill” strategy [40]. In this case, virus transcription, protein expression, and virion production is induced by latency reversing agents, which is followed by activation of immune-mediated clearance. Therapeutic vaccines such as CaP NPs could increase the quantity and efficacy of virus-specific cytotoxic CD8^+^ T cell responses to enhance the killing of latently infected cells [41,42]. With the flexible CaP NP system, it would be possible to individually prime antigen specific T cell responses against vulnerable regions of the virus, which has been effective for HIV control [43,44], but may also be important for the reinforcement of tumor-specific immune responses [45].

In conclusion, in the current study, we present a reliable in vitro approach to mature DCs and to activate and expand functional human CD8^+^ T cells specifically for a viral antigen. TLR ligands improve the maturation of DCs and, thus, influence the priming of T cells. Encapsulation of biomolecules into CaP NP carriers strongly enhances this effect by inducing higher expression of costimulatory molecules and proinflammatory cytokines. Our results confirm our previous findings from mouse models in a human context, which suggests CaP NPs are a potential vaccine system against viral infections or cancer.

## 5. Conclusions

Immunosuppressive mechanisms such as T cell exhaustion facilitate the need for new immunotherapeutic strategies to enhance immune activation during therapeutic vaccination. Nanoparticle-based carrier systems can present pathogen mimetic abilities by antigen delivery and by targeting and activating antigen presenting cells. Therapeutic vaccines could therefore increase the quantity and efficacy of antigen-specific CD8^+^ T cell responses to enhance the killing of latently infected or cancerous cells. Here we present a reliable ***in vitro*** approach to expand functional human virus-specific CD8^+^ T cells by antigen and TLR ligands functionalized CaP nanoparticles. DC maturation and activation of CD8^+^ T cells was more increased compared to the treatment with soluble agents suggesting CaP nanoparticles as a powerful strategy for potent therapeutic vaccination approaches against viral infections or cancer.

## Figures and Tables

**Figure 1 vaccines-08-00110-f001:**
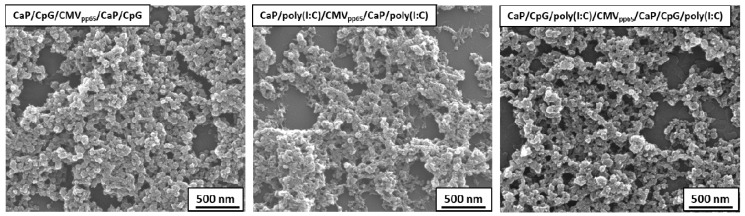
Scanning electron micrographs of functionalized calcium phosphate (CaP) nanoparticles. Representative scanning electron microscopy (SEM) images of triple shell CaP nanoparticles (NPs) are shown containing Toll-like receptor (TLR) ligands and CMVpp65 peptide. The diameter of the mostly spherical particles was 60–75 nm.

**Figure 2 vaccines-08-00110-f002:**
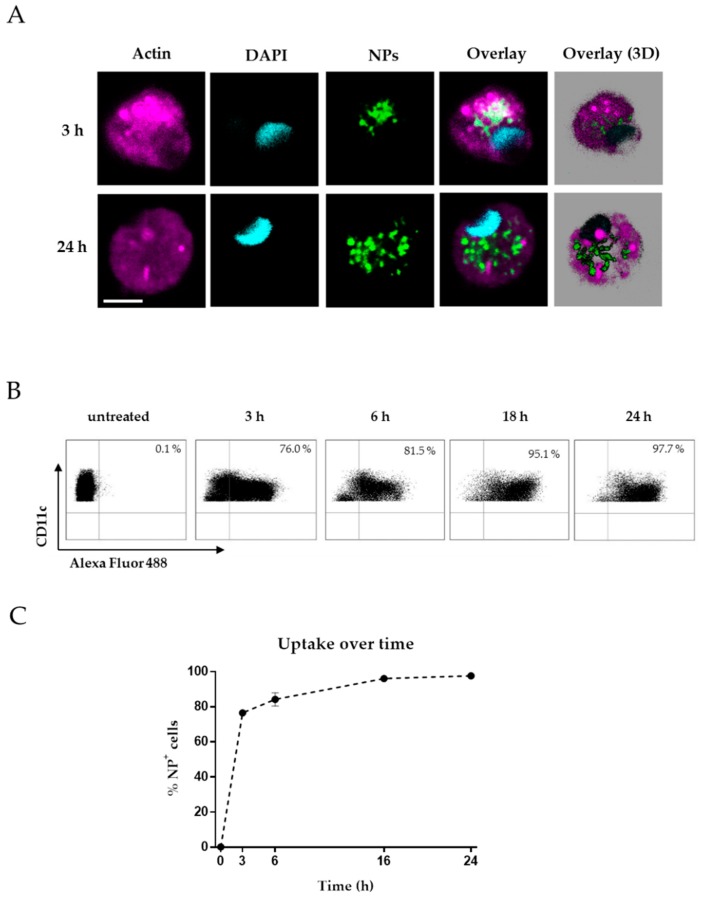
Uptake of functionalized CaP nanoparticles by dendritic cells (DCs). (**A**) Confocal laser scanning microscopic images demonstrating the uptake of green fluorescing CpG-Alexa488 labeled functionalized CaP nanoparticles by human cDCs after 3 h and 24 h incubation. The reconstruction of all z-stacks (30 slices, 0.3 μm thick) in a 3D representation. Magenta: actin staining. Blue: nuclear staining by DAPI. Green: nanoparticles. Scale bars 5 μm. (**B**–**C**) PBMCs were isolated from peripheral blood and incubated with CpG-Alexa488 labeled functionalized CaP nanoparticles. After 3, 6, 16, and 24 h PBMCs were analyzed for CD11c^+^ CD14^−^ Alexa488^+^ cells. Bars represent means with SD. The results of three independent experiments are shown.

**Figure 3 vaccines-08-00110-f003:**
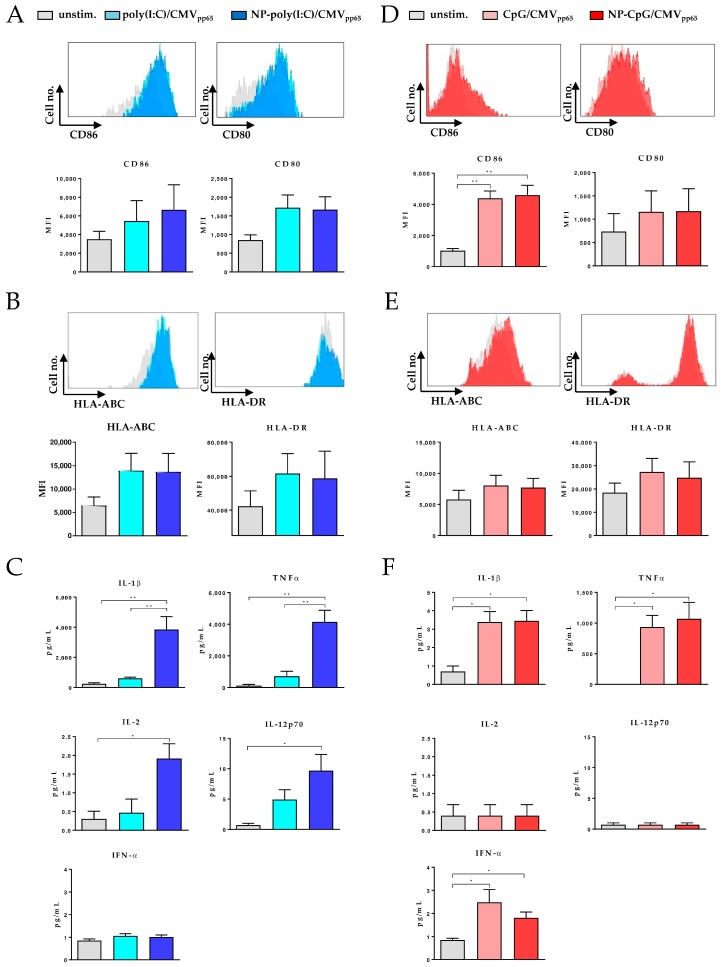
Maturation of conventional DCs (cDCs) and plasmacytoid DCs (pDCs) by functionalized CaP nanoparticles. (**A**) Representative histograms show expression of costimulatory molecules cDCs. Blood mononuclear cells (PBMC) isolated cDCs were stimulated with poly(I:C) functionalized CaP NPs or soluble poly(I:C). After 24 h expression of costimulatory molecules, CD86 and CD80 was analyzed by flow cytometry. The mean fluorescence intensity (MFI) is depicted. (**B**) 24 h after stimulation expression of MHC I (HLA-ABC) and MHC II (HLA-DR) was analyzed by flow cytometry. (**C**) Supernatants from stimulated cDCs were analyzed for proinflammatory cytokines after 24 h. (**D**) PBMC-isolated pDCs were stimulated with CpG functionalized CaP NPs or soluble CpG. After 24 h expression of costimulatory molecules, CD86 and CD80 was analyzed by flow cytometry. (**E**) MHC I (HLA-ABC) and MHC II (HLA-DR) expression was analyzed by flow cytometry. (**F**) Supernatants from stimulated pDCs were investigated for proinflammatory cytokines after 24 h of incubation. Results are summarized from at least three independent experiments. Bars represent means with SEM. One-way ANOVA was used, which was followed by the Tukey’s multiple comparisons test. * *p* < 0.05, ** *p* < 0.01.

**Figure 4 vaccines-08-00110-f004:**
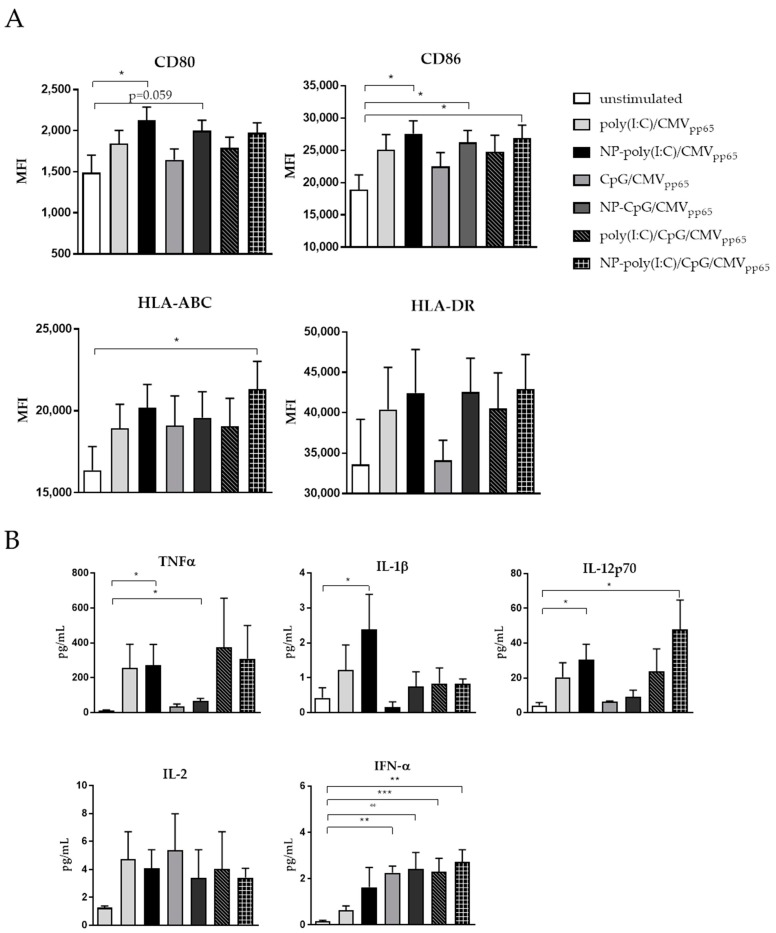
Maturation of monocyte-derived DCs (mDCs) by functionalized CaP nanoparticles. (**A**) mDCs were stimulated with poly(I:C), CpG or poly(I:C), and CpG functionalized CaP NPs or soluble ligands. The expression of the costimulatory molecules CD86, CD80, HLA-ABC, and HLA-DR was analyzed 24 h after stimulation by flow cytometry. (**B**) Supernatants from stimulated mDCs were investigated for proinflammatory cytokines after 24 h. Results are summarized from at least four independent experiments. Bars represent means with SEM. One-way ANOVA was used when followed by the Tukey’s multiple comparisons test. * *p* < 0.05. ** *p* < 0.01. *** *p* < 0.001.

**Figure 5 vaccines-08-00110-f005:**
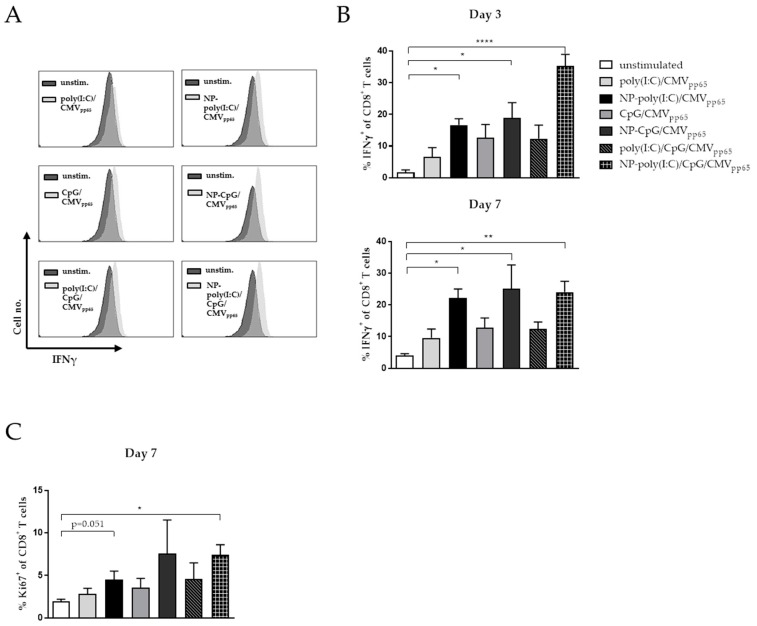
Induction of cytomegalovirus (CMV)-specific CD8^+^ T cell response by functionalized CaP nanoparticle-stimulated mDCs. (**A**) Representative histograms of flow cytometry analysis of IFNγ producing CD8^+^ T cells after three days are depicted. (**B**) Untreated or treated mDCs were co-cultured with CD8^+^ T cells isolated from CMV-positive patients. Frequencies of IFNγ producing CD8^+^ T cells were analyzed after three days or seven days of co-culture by flow cytometry. Coculture experiments were performed six times. Each experiment represents a different subject. (**C**) Proliferation of CD8^+^ T cells was analyzed by Ki67 expression. Results from three independent experiments are depicted. Each experiment represents a different donor. Bars represent means with SEM. One-way ANOVA was used followed by the Tukey’s multiple comparisons test. * *p* < 0.05. ** *p* < 0.01. **** *p* < 0.0001.

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
