# Peer review of "Effective Activation of Human Antigen-Presenting Cells and Cytotoxic CD8+ T Cells by a Calcium Phosphate-Based Nanoparticle Vaccine Delivery System"

_vaccines, 2020, doi:10.3390/vaccines8010110_

Round 1

Reviewer 1 Report

This manuscript presents the use of CaP nanoparticles as viral vaccine carriers. The manuscript is well written in English, organized and only minor mistakes should be addressed:

Introduction:

I will rephrase this: Due to their small size NPs are predestined to target and to be 42 engulfed by antigen presenting cells (APC) such as dendritic cells (DC) or macrophages [5]. That’s not completely true. NPs (<100nm) are more likely to go through the lymphatic system to the lymph nodes, and sub-micron particles are more likely to be uptake by APC (and even the size here plays an important role on which cell will engulf the particle. I will also include on the introduction the advantages if using CaP NPs rather than other polymeric materials on the introduction. And more references can be added about the use of CaP as vaccine carriers to support its use.

Materials and Methods

Synthesis of functionalized CaP nanoparticles: I don’t undertand how the NPs were prepared. It is clear how they make the one shell NPs, but it should be clarified how they make the triple shell ones. How was the number of particles per mL calculated?

Results

The size, PDI, zeta potential and composition of the NPs should be included in results not in methods, and it will be appropriate to include the SEM image of the NPs. On the manuscript.

Author Response

Introduction:

I will rephrase this: Due to their small size NPs are predestined to target and to be 42 engulfed by antigen presenting cells (APC) such as dendritic cells (DC) or macrophages [5]. That’s not completely true. NPs (<100nm) are more likely to go through the lymphatic system to the lymph nodes, and sub-micron particles are more likely to be uptake by APC (and even the size here plays an important role on which cell will engulf the particle. I will also include on the introduction the advantages if using CaP NPs rather than other polymeric materials on the introduction. And more references can be added about the use of CaP as vaccine carriers to support its use.”

Authors’ response: We agree with the reviewer and added additional information on the uptake of nanoparticles and the use of CaP nanoparticles as successful vaccine carrier to the introduction of the revised version of our manuscript.

Materials and Methods

Synthesis of functionalized CaP nanoparticles: I don’t understand how the NPs were prepared. It is clear how they make the one shell NPs, but it should be clarified how they make the triple shell ones. How was the number of particles per mL calculated?”

Authors’ response: We apologize for the missing information. We extended the methods on the synthesis of triple shell nanoparticles and described how the particle number was calculated in the revised version of the manuscript.

Results

The size, PDI, zeta potential and composition of the NPs should be included in results not in methods, and it will be appropriate to include the SEM image of the NPs. On the manuscript.”

Authors’ response: This is a good idea. We included a detailed characterization of CaP nanoparticles into the results section. We also added SEM images of the CaP nanoparticles to illustrate their shape and size.

Reviewer 2 Report

Brief summary:

The manuscript of Scheffel et al. reports the development of a calcium phosphate nanoparticle (NP) delivery system loaded with poly(I:C), CpG and a CMV-derived peptide with the goal to activate human antigen-presenting cells for vaccination purposes. These Poly(I:C)/CMV NPs and CpG/CVM NPs were able to activate cDCs, pDCs and mDCs and induce the production of proinflammatory cytokines from CMV-positive human PBMCs. Lastly, it was shown that mDCs primed with Poly(I:C)/CMV NPs and CpG/CVM NPs were able to induce IFN-g production in CD8+ T cells from patients infected with CMV. The results suggest that the developed NPs have potential to be used as vaccine candidates against infectious diseases or cancer in which APC activation and CD8+ T cells priming is important for protection.

Major comments:

In general, the introduction and the materials & methods section are well written. However, the peptide-pulsed CD8+ T cells mentioned in the M&M section, do not seem to be mentioned further in the manuscript. Can you explain what these were used for?

Section 3.1 This is a great experiment to confirm that the NPs are internalized by the DCs. In this experiment I am missing dosing of the NP formulation which was added to the cells. It is important to know the number of NPs that were added relative to the number of cells. Ideally this ratio (NP/Cells) should be mentioned for each experiment.

Section 3.2: This experiment is well designed and showed clearly that there is increased activation and production of proinflammatory cytokines for Poly(I:C) encapsulated in the NP. It would be interesting to discuss why there is no difference observed between soluble CpG + CMVpp65 and NP-CpG/CMVpp65 in the pDCs. Does CpG get endocytosed effectively in soluble form and is it therefore easy to compete with CpG-NPs? Is there a concentration issue? Or is there something else that might explain this? I suggest to clarify to the reader that your pDCs express TLR9 and not TLR3 and therefore you are only using CpG particles with pDCs in Fig 2. Visa versa, make a note that cDCs express TLR3 and not TLR9 and therefore you are only using poly I:C particles with cDCs in Fig 2. This info can be extracted from lines 51 and lines 211, but, in my opinion,  it will be more clear if this is mentioned in section 3.2.

Figure 3: It would be nice to show the results for IL-2, since it is an important cytokine for CD8+ T cell activation.

Figure 4 shows the capability of the NP-CpG and NP-Poly(I:C) together with the CMV peptide to induce CMV-specific CD8+ T cells responses. However, in line 32 and line 269 it is stated that this figure shows the expansion of CMV-specific T cells. In my opinion, the expansion is not shown here, because we are not looking at the number of T cells, or proliferation markers, but at the T cells that are secreting IFN-g. The amount of CD8+ T cells secreting IFN-g could have gone up while the total number of CD8+ T cells stayed the same, and the cells might have not proliferated as well. In my opinion it is very important that the right CD8+ T cell markers are used in order to make the statements made in the article correct.

Section 4: In the discussion in line 303 it is mentioned that besides IL-12 and TNF-a, IL-2 is very important for CD8+ T cell activation. It`s unclear in the article why IL-2 results are only shown in Fig. 2 for the cDCs. I would also show IL-2 results for the pDCs and the mDCs to strengthen the conclusions made in the discussion.

Minor comments:

Line 132: put centrifugation not in rpm but in g.

Line 137: add “rh” to IL-4.

Line 145: is 100 g/mL pen/strep correct? Earlier in section “Generation of monocyte derived DCs” pen/strep was mentioned as 5%.

Line 169: Was it confirmed that the cells are effector CD8+ T cells, and not other kinds of T cells?

Line 177-179: Markers CD83, HLA-DR, HLA-A,B,C and Gzmb were not mentioned in the results. Are these markers not tested for or is there supplementary or unpublished data?

Line 198: Alexa488-labelled CaP NPs should be CpG-Alexa488-labelled Cap NP, correct?

Line 297-300: The statement that the secretion of proinflammatory cytokines such as IL-12, IL-2 and TNF-a was increased after CA NPs treatment is not completely correct. IL-2 results are only shown for cDCs stimulated with Poly(I:C) in fig 2, so it is unknown if pDCs and mDCs produced IL-2 after stimulation.

Author Response

Major comments:

In general, the introduction and the materials & methods section are well written. However, the peptide-pulsed CD8+ T cells mentioned in the M&M section, do not seem to be mentioned further in the manuscript. Can you explain what these were used for?

Authors’ response: Peptide pulsed PBMCs were used for the restimulation of Ag-specific CD8+ T cells from CMV+ patients. We apologize for the misunderstanding and partially rephrased the description for this part in the MM section.

“Section 3.1 This is a great experiment to confirm that the NPs are internalized by the DCs. In this experiment I am missing dosing of the NP formulation which was added to the cells. It is important to know the number of NPs that were added relative to the number of cells. Ideally this ratio (NP/Cells) should be mentioned for each experiment.”

Authors’ response: We absolutely agree with the reviewer and added the NP dose for the stimulation of DCs to the revised manuscript. In addition, dosing information were included into the MM section Stimulation of DCs with functionalized CaP nanoparticles.

“Section 3.2: This experiment is well designed and showed clearly that there is increased activation and production of proinflammatory cytokines for Poly(I:C) encapsulated in the NP. It would be interesting to discuss why there is no difference observed between soluble CpG + CMVpp65 and NP-CpG/CMVpp65 in the pDCs. Does CpG get endocytosed effectively in soluble form and is it therefore easy to compete with CpG-NPs? Is there a concentration issue? Or is there something else that might explain this? I suggest to clarify to the reader that your pDCs express TLR9 and not TLR3 and therefore you are only using CpG particles with pDCs in Fig 2. Visa versa, make a note that cDCs express TLR3 and not TLR9 and therefore you are only using poly I:C particles with cDCs in Fig 2. This info can be extracted from lines 51 and lines 211, but, in my opinion,  it will be more clear if this is mentioned in section 3.2. “

Authors’ response: Interestingly, we did not observe differences in the expression levels of surface molecules after NP stimulation of DCs compared to stimulation with soluble TLR ligands. This could be due to the time point of analysis and the fact, that the concentration of the TLR ligands is adapted to the concentration used in murine in vitro studies. Using this concentration could obscure the advantageous NP effect in vitro. We have included this into the Discussion section. In addition, we added more information about the expression of TLRs on pDCs and cDCs to justify the use of poly(I:C) or CpG.

 “Figure 3: It would be nice to show the results for IL-2, since it is an important cytokine for CD8+ T cell activation“

Authors’ response: We agree that this is of great interest and included analysis of IL-2 concentrations in the supernatants of DCs. 

“Figure 4 shows the capability of the NP-CpG and NP-Poly(I:C) together with the CMV peptide to induce CMV-specific CD8+ T cells responses. However, in line 32 and line 269 it is stated that this figure shows the expansion of CMV-specific T cells. In my opinion, the expansion is not shown here, because we are not looking at the number of T cells, or proliferation markers, but at the T cells that are secreting IFN-g. The amount of CD8+ T cells secreting IFN-g could have gone up while the total number of CD8+ T cells stayed the same, and the cells might have not proliferated as well. In my opinion it is very important that the right CD8+ T cell markers are used in order to make the statements made in the article correct.”

 Authors’ response: Antigen-specific restimulation is a regular procedure to identify virus-specific T cells via IFNγ production. Since the coculture of untreated DCs with CD8+ T cells did not enhance the numbers of IFNγ producing CD8+ T cells compared to all other coculture conditions, we are convinced that this experiment properly demonstrates the activation and expansion of CD8+ T cells. To investigate proliferation of CD8+ T cells, we performed Ki67 staining on day 7 of coculture. Of note, we observed proliferation of CD8+ T cells, but the percentages of Ki67+ cells within this CD8+ T cell pool is low due to the fact that the majority of the T cells in the coculture is still unspecific for the CMVpp65 peptide.

“Section 4: In the discussion in line 303 it is mentioned that besides IL-12 and TNF-a, IL-2 is very important for CD8+ T cell activation. It`s unclear in the article why IL-2 results are only shown in Fig. 2 for the cDCs. I would also show IL-2 results for the pDCs and the mDCs to strengthen the conclusions made in the discussion.”

Authors’ response: We agree that this is an important information. However, IL-2 and IL-12 secretion were not detectable by pDCs in our experiments. Therefore, we first decided to resign their presentation in the manuscript. We now have included the results for IL-2 and IL-12 in the manuscript and hope that this will answer upcoming questions.

Minor comments:

Line 132: put centrifugation not in rpm but in g.”

Authors’ response:We changed rpm to g here.

“Line 137: add “rh” to IL-4.”

 Authors’ response: Now we use the term rhIL-4 here.

“Line 145: is 100 g/mL pen/strep correct? Earlier in section “Generation of monocyte derived DCs” pen/strep was mentioned as 5%.”

Authors’ response: We matched the two descriptions to avoid confusions.

“Line 169: Was it confirmed that the cells are effector CD8+ T cells, and not other kinds of T cells?”

Authors’ response: We did not confirm the effector cell status of the cells prior to coculture. It is also not important whether these are effector cells or not. We rephrased this description.

“Line 177-179: Markers CD83, HLA-DR, HLA-A,B,C and Gzmb were not mentioned in the results. Are these markers not tested for or is there supplementary or unpublished data?”

 Authors’ response: We apologize for this mistake. We excluded CD83 and GzmB from the text.

“Line 198: Alexa488-labelled CaP NPs should be CpG-Alexa488-labelled Cap NP, correct?”

Authors’ response: CpG-Alexa488-labelled Cap NP is the right annotation. We corrected this in the revised version of the manuscript.

“Line 297-300: The statement that the secretion of proinflammatory cytokines such as IL-12, IL-2 and TNF-a was increased after CA NPs treatment is not completely correct. IL-2 results are only shown for cDCs stimulated with Poly(I:C) in fig 2, so it is unknown if pDCs and mDCs produced IL-2 after stimulation.”

Authors’ response: We now have included the results for IL-2 and IL-12 in the manuscript and hope that this will answer upcoming questions (see above).

Reviewer 3 Report

In this study, the authors demonstrated that TLR ligand-functionalized calcium phosphate-based nanoparticles (Cap NP) increased activation of human DCs as compared to soluble TLR ligands. The authors also showed that these DCs had a higher ability to induce CD8+ T cell response. The work described would be is of interest to the field, contributing to the development of more efficient vaccines. However, several concerns should be addressed before publication as following.

Major points

In Figure 1, the flow cytometry result shown is insufficient to support the claim that CpG Alexa488 is uptake by DCs. Authors should perform additional experiments, ex) confocal microscopic analysis. In Figure 2, since TCR-antigen-MHC interaction is critical for T cell activation, authors should show the expression level of HLA molecules on DCs. Also, the production level of type I interferons should be analyzed as they were known to be important cytokines for CaP NPs-mediated DC activation in the authors’ previous studies. In Figure 3, NP-poly(I:C)/CMVpp65 did not statistically increase expression of CD80, CD86, TNFalpha, IL-1beta, and IL-12 as compared to poly(I:C)/CMVpp65. Similarly, neither NP-CpG/CMVpp65 nor NP-poly(I:c)/CpG/CMVpp65 statistically significantly increased the expression of those molecules on DCs. In Figure 4, the representative FACS plots for ‘%IFNg of CD8 T cells’ should be provided. Also, the authors should perform additional experiments for T cell expansion, for example, Ki-67 expression and CFSE staining. It is unclear whether data are the representative results of independent experiments. Also, the authors should indicate the total number of human subjects used for this study. It would be informative if the authors discuss the role of Cap NP in DCs to activate CD4 cells.

Minor points

Abbreviations should be defined at first use in the abstract and again in the body of the manuscript. For example, in line 31, “Nanoparticle-based carrier…..” should be changed to “Nanoparticle (NP)-based carrier…”. In lines 175-280, I do not find any data where anti-CD14, CD83, HLA-DR, GzmB, Foxp3, HLA-A, B, C antibodies were used. Please provide a more detailed method for cytokine quantification in Materials and Methods. In Figure 1, please provide FACS plots for 6 and 24 hrs as well. CpG treatment is known to increase CD80 expression on human pDCs in many other studies, which is different from the data in Figure 2. In lines 224-226, “Stimulation of pDCs with CpG functionalized ~~~~~~~soluble TLR ligand” needs to be rephrased.

Author Response

“In Figure 1, the flow cytometry result shown is insufficient to support the claim that CpG Alexa488 is uptake by DCs. Authors should perform additional experiments, ex) confocal microscopic analysis.”

Authors’ response: To further analyse the uptake by DCs we additionally performed CLSM studies with CpG-Alexa488-labelled Cap NPs. Here, we clearly demonstrate the presence of NPs inside the cells.

“In Figure 2, since TCR-antigen-MHC interaction is critical for T cell activation, authors should show the expression level of HLA molecules on DCs”

Authors’ response: We included the expression levels of MHC I and MHC II molecules on DCs in the revised manuscript. The expression of both molecules was enhanced after stimulation. We observed a significantly increased expression of MHC I on mDCs after NP-poly(I:C)/CpG/CMV stimulation. Importantly, at the same time we observed the highest CD8+ T cell response in DC-T cell coculture experiments, where mDCs were stimulated with NP-poly(I:C)/CpG/CMV in adavance (Figure 5).

“Also, the production level of type I interferons should be analyzed as they were known to be important cytokines for CaP NPs-mediated DC activation in the authors’ previous studies.”

Authors’ response: We agree that this is of high interest. We included analysis of IFN-α level in supernatants of DC cultures. IFN-α is secreted by pDCs and mDCs after NP treatment.

“In Figure 3, NP-poly(I:C)/CMVpp65 did not statistically increase expression of CD80, CD86, TNFalpha, IL-1beta, and IL-12 as compared to poly(I:C)/CMVpp65. Similarly, neither NP-CpG/CMVpp65 nor NP-poly(I:c)/CpG/CMVpp65 statistically significantly increased the expression of those molecules on DCs.”

Authors’ response: Interestingly, we did not observe differences in the expression levels of surface molecules after NP stimulation of DCs compared to stimulation with soluble TLR ligands. This could be due to the time point of analysis and the fact, that the concentration of the TLR ligands is adapted to the concentration used in murine in vitro studies This could obscure the advantageous NP effect in vitro. We have included this into the Discussion section. In addition, we added more information about the expression of TLRs on pDCs and cDCs to justify the use of poly(I:C) or CpG.

“In Figure 4, the representative FACS plots for ‘%IFNg of CD8 T cells’ should be provided. Also, the authors should perform additional experiments for T cell expansion, for example, Ki-67 expression and CFSE staining. It is unclear whether data are the representative results of independent experiments. Also, the authors should indicate the total number of human subjects used for this study.”

Authors’ response: We have included representative FACS histograms for day 3 of IFNγ+ CD8+ T cells. Experiments were performed six times. In each experiment a different subject was used. We have added this important information within the MM section and figure legend.

Antigen-specific restimulation is a regular procedure to identify virus-specific T cells via IFNγ production. Since the coculture of untreated DCs with CD8+ T cells did not enhance the numbers of IFNγ producing CD8+ T cells compared to all other coculture conditions, we are convinced that this experiment properly demonstrates the activation and expansion of CD8+ T cells. To investigate proliferation of CD8+ T cells, we performed Ki67 staining on day 7 of coculture. Of note, we observed proliferation of CD8+ T cells, but the percentages of Ki67+ cells within this CD8+ T cell pool is low due to the fact that the majority of the T cells in the coculture is still unspecific for the CMVpp65 peptide.

“It would be informative if the authors discuss the role of Cap NP in DCs to activate CD4 cells.”

Authors’ response: In previous publications we have demonstrated the encapsulation of poly(I:C) or/and CpG together with a MHC class II restricted model antigen for the activation of CD4+ T cells (Sokolova et al. 2010, Knuschke et al. 2013). We have included this point into the Discussion section. However, in the current work we have focused on CD8+ T cells, since they are the main  targets of immunotherapy during cancer or chronic viral infection. Induction of CD4+ T cell responses might also be enhanced by CaP NP loaded human DCs.

“Abbreviations should be defined at first use in the abstract and again in the body of the manuscript. For example, in line 31, “Nanoparticle-based carrier…..” should be changed to “Nanoparticle (NP)-based carrier…”.”

Authors’ response: Thank you for this advice. We have corrected the abbreviation for NP.

“In lines 175-280, I do not find any data where anti-CD14, CD83, HLA-DR, GzmB, Foxp3, HLA-A, B, C antibodies were used.”

Authors’ response: We apologize for the confusion. CD14 was used for the gating of DCs. CD83, GzmB and Foxp3 were not analyzed in this study. We have corrected this in the revised manuscript.

“Please provide a more detailed method for cytokine quantification in Materials and Methods.”

Authors’ response: We extended the description of the Luminex method to detect cytokines in cell supernatants. We hope that this will explain this method more precisely.

“In Figure 1, please provide FACS plots for 6 and 24 hrs as well.”

Authors’ response: We added additional dot plots demonstrating the uptake of NPs in the revised manuscript.

“CpG treatment is known to increase CD80 expression on human pDCs in many other studies, which is different from the data in Figure 2.”

Authors’ response: We performed two additional experiments to further increase the sample number. Although differences are not significant, we do see now a slight enhancement of CD80 on pDCs.

“In lines 224-226, “Stimulation of pDCs with CpG functionalized ~~~~~~~soluble TLR ligand” needs to be rephrased.”

Authors’ response: We rephrased this sentence in the revised version of the manuscript.

Round 2

Reviewer 1 Report

Authors have corrected  the submitted manuscript and all my comments have been successfully addressed. Thus, it can be published in the present form. 

Reviewer 2 Report

Thank you for your changes to the manuscript. You have addressed all the comments that I had, and I agree with the edits you made in the revised manuscript. In my opinion, this is now ready for publication.

Reviewer 3 Report

This revised article adequately addressed most of the reviewers’ comments, which greatly increased the quality of the manuscript.